# Technical note: Further adjustments to the Rock-Eval® thermal analysis for soil organic and inorganic carbon quantification to avoid post-hoc corrections

Joséphine Hazera<sup>1,2,7</sup>, David Sebag<sup>2</sup>, Isabelle Kowalewski<sup>2</sup>, Herman Ravelojaona<sup>2</sup>, Eric Verrecchia<sup>3</sup>, Gergely Jakab<sup>4</sup>, Dóra Zacháry<sup>4</sup>, Florian Schneider<sup>5</sup>, Luca Trombino<sup>6</sup>, Raphaël J. Manlay<sup>7,8</sup>, Julien Fouché<sup>9</sup> and Tiphaine Chevallier<sup>7</sup>

<sup>1</sup>Institut des Sciences de la Terre de Paris ISTeP – UMR 7193, Sorbonne Université, CNRS, Paris, 75005, France

<sup>2</sup>Direction Sciences de la Terre et Technologies de l'Environnement, IFP Énergies Nouvelles, Rueil-Malmaison, 92852, France

<sup>3</sup>Institute of Earth Surface Dynamics, University of Lausanne, Lausanne, 1015, Switzerland

<sup>4</sup>HUN-REN Research Centre for Astronomy and Earth Sciences, Geographical Institute, Budapest, 1121, Hungary

<sup>5</sup>Institute of Climate-Smart Agriculture, Thünen, Braunschweig, 38116, Germany

<sup>6</sup>Earth Science Department, University of Milan, Milan, 20133, Italy

<sup>7</sup>Eco&Sols, University of Montpellier, CIRAD, Institut Agro Montpellier, INRAE, IRD, Montpellier, 34060, France <sup>8</sup>AgroParisTech, Palaiseau, 91120, France

<sup>9</sup>Lisah, University of Montpellier, Institut Agro Montpellier, INRAE, IRD, Montpellier, 34060, France

Correspondence to: Joséphine Hazera (josephinehazera@gmail.com)

Abstract. Accurate quantifications of soil organic (SOC) and inorganic (SIC) carbon are essential for a better understanding of the global carbon cycle. The procedures usually used to quantify SOC and SIC (e.g., elemental analysis after pretreatment) rely on various approximations and can lead to analytical errors. Ramped thermal analyses are increasingly investigated to quantify SOC and SIC by heating a single aliquot and continuously measuring the carbonaceous compounds emitted. The Rock-Eval® thermal analysis (RE) has been standardized to estimate organic and inorganic C contents of oil-bearing rocks through two parameters named TOC and MINC, respectively. Moreover, its pyrolysis phase before the oxidation provides the basis for calculating indices to characterize soil organic matter (SOM). However, statistical post-hoc corrections of TOC and MINC are needed to adjust their estimations of SOC and SIC contents because SOC and SIC decomposition signals overlap at the end of the pyrolysis. A new cycle with a final pyrolysis temperature of 520 °C (PYRO520) instead of 650°C is investigated to avoid SIC decomposition while preserving OM characterization during pyrolysis. The results are compared to the quantifications obtained with the standard analysis cycle (PYRO650) and by elemental analysis after pretreatments. The PYRO520 cycle corrects the misallocation of the end-of-pyrolysis signals between the TOC and MINC parameters and thus accurately and repeatably estimated SOC and SIC contents measured by EA after pretreatments without needing post-hoc corrections. Moreover, the values and interpretations of the indices characterizing SOM are not drastically modified by the pyrolysis modification.

Ebouel et al., 2024; Shamrikova et al., 2024).

#### 1 Introduction

Two carbon (C) forms can be found in soils: soil organic carbon (SOC) and soil inorganic carbon (SIC). Since the early 1990s, only about 4% of soil C publications have addressed SIC (Raza et al., 2024). Traditionally, SIC has been viewed as inert over a human lifetime, only slightly affected by land use and agricultural practices, and thus considered irrelevant to studies on soil C sequestration and soil ecology (Dina Ebouel et al., 2024; Raza et al., 2024). However, an increasing number of studies shows that the presence of SIC can enhance SOC stabilization (Rowley et al., 2018; Qafoku et al., 2023; Shabtai et al., 2023) and may act either as a sink (Entry et al., 2004; Manning, 2008; Cailleau et al., 2011; Wang et al., 2015; Bughio et 40 al., 2016; Raheb et al., 2017; Gao et al., 2017; Liu et al., 2018; Gatz-Miller et al., 2022; Liu et al., 2023) or as a source of atmospheric CO<sub>2</sub> (Emmerich, 2003; Bertrand et al., 2007; Chevallier et al., 2016; Raza et al., 2021; Zamanian et al., 2021; Guo et al., 2021; Zhao et al., 2022). Despite these findings, the influence of SIC-related processes on the global C cycle remains uncertain. Consequently, accurate quantifications of SOC and SIC are essential for completing soil C databases, improving digital mapping of SIC, and integrating SIC dynamics into global C cycle models (Sharififar et al., 2023; Dina 45 Ebouel et al., 2024; Raza et al., 2024). The elemental analysis (EA) remains the reference method for soil C quantification. It consists in high-temperature combustion (> 1000 °C) of the soil sample and measurement of the emitted carbon dioxide (CO<sub>2</sub>; ISO, 1995; Bispo et al., 2017; Apesteguia et al., 2018; Shamrikova et al., 2023). Both SOC and SIC decompose at this temperature. To quantify SOC and SIC, the total carbon (TC) is first estimated by EA on one aliquot, and then either SOC or SIC is estimated on a second aliquot by removing the unwanted C form prior to EA or by using specific methods quantifying SOC (e.g., wet oxidation) or SIC (e.g., calcimetry). However, SIC removal by acidification or SOC removal by combustion can be incomplete or even alter the untargeted C form (Schlacher and Connolly, 2014; Apesteguia et al., 2018; Shamrikova et al., 2023, 2024). Moreover, specific quantification methods depend on approximations for estimating SOC yields from wet oxidation, while calcimetry is calibrated exclusively for quantifying calcium carbonate and thus misses other carbonate forms (e.g., calcium 55 magnesium carbonate) (Apesteguia et al., 2018; Nayak et al., 2019; Shamrikova et al., 2023, 2024). The unquantified C form is then calculated using the difference (e.g., SIC = TC - SOC), leading to an increase in the measurement error of the result (e.g.,  $\Delta SIC = \sqrt{\Delta TC^2 + \Delta SOC^2}$ ). Sample heterogeneity and/or very low or high contents of one C form can also lead to inconsistencies and even negative or null values (Apesteguia et al., 2018; Shamrikova et al., 2023, 2024). The loss-onignition method quantifies SOC and SIC by heating a single aliquot and by measuring the mass loss during the heating. The 60 mass loss up to 450-600 °C is usually attributed to organic matter (OM) decomposition and the mass loss up to 850-950 °C to carbonate decomposition. However, the temperature limits reported in the literature are variable, and other processes (e.g., clay dehydroxylation) can contribute to mass loss. Moreover, SOC contents are estimated by assuming that OM contains on average 58% carbon, a proportion that can vary widely between soils (from 40% to 71%; Apesteguia et al., 2018; Dina




Thermal analyses are increasingly investigated to quantify both SOC and SIC on a single aliquot by continuously measuring the carbonaceous compounds released either during oxidation alone (Vuong et al., 2013, 2015; Apesteguia et al., 2018) or during pyrolysis and oxidation (Delahaie et al., 2023; Hazera et al., 2023; Koorneef et al., 2023). Since SIC decomposes at higher temperatures than SOC, these methods use temperature boundaries to distinguish the compounds emitted by the SOC decomposition from those emitted by the SIC decomposition. Among these methods, the Rock-Eval® thermal analysis (RE) has been standardized to estimate organic and inorganic C contents in oil-bearing rocks through two parameters named TOC and MINC, respectively. The advantage of RE is to include a pyrolysis phase before the oxidation during which hydrocarbon compounds (HC) are measured in addition to carbon monoxide (CO) and CO<sub>2</sub> (Lafargue et al., 1998; Behar et al., 2001). The HC measurement is used to calculate indices that are increasingly used in soil science to characterize soil OM (Disnar et al., 2003; Carrie et al., 2012; Sebag et al., 2016; Cécillon et al., 2018; Malou et al., 2023).

Nevertheless, thermal analyses are not exempt from uncertainties on temperature boundaries to be used for distinguishing the signals from SOC and SIC thermal decompositions. Recent quantifications of SOC and SIC by RE in calcareous samples showed that the sum of TOC and MINC parameters accurately estimates the TC content, while the TOC parameter underestimates the SOC content and the MINC parameter overestimates the SIC content. A proportion of SOC is accounted in the MINC parameter because of an overlap of the SOC and SIC decomposition signals at the end of the pyrolysis (Delahaie et al., 2023; Hazera et al., 2023; Koorneef et al., 2023). Indeed, SIC thermal decomposition begins at the end of the pyrolysis, while a proportion of SOC remains in the crucible. The CO2 emitted by SIC decomposition reacts with the residual organic C to form two CO molecules (Boudouard's reaction:  $CO_2 + Corg \rightarrow 2CO$ ). The production of CO at the end of pyrolysis (i) leads to the arbitrary halving of the CO signal between TOC and MINC parameters although the yield of the Boudouard's equilibrium depends on temperature, and (ii) increases the uncertainties on temperature boundaries to be used to distinguish SOC and SIC signals since SIC thermal decomposition usually does not produce CO (CaCO<sub>3</sub>  $\xrightarrow{\Delta}$  CaO + CO<sub>2</sub>). Thus, distinguishing SOC and SIC decomposition signals during pyrolysis remains uncertain and likely depends on the forms of SOC and SIC in the soil sample. A post-hoc correction has been proposed consisting in subtracting a proportion of the TOC parameter (8-9%) from the MINC parameter (Sebag et al., 2022a; Hazera et al., 2023) and adding it to the TOC parameter (Disnar et al., 2003). However, this correction may not be suitable for all soil types and remains calibrated on SOC and SIC contents estimated by the reference methods, which themselves include measurement errors.

Preventing SIC thermal decomposition by reducing the final temperature of pyrolysis could avoid the need for post-hoc corrections for SOC and SIC quantification. The standard pyrolysis reaches 650 °C, a high temperature originally chosen to measure the HC emitted by the thermal decomposition of the most mature organic matters of oil-bearing rocks (Lafargue et al., 1998). However, in soils, the most persistent OM are hydrogen-depleted so their decomposition emit few HC at high temperatures (Disnar et al., 2003; Barré et al., 2016; Sebag et al., 2016). The HC emitted by soil samples at the end of the pyrolysis are rare and not relevant for the characterization of soil OM (Pacini et al., 2023; Deluz et al., 2024). Delarue et al. (2013) even suggested that the HC emitted at the end of the pyrolysis could be mainly products from secondary reactions






occurring in the pyrolysis furnace. Therefore, lowering the final temperature of pyrolysis should not compromise the characterization of soil OM through the HC-based indices. During pyrolysis, the thermal decomposition of common soil carbonates (e.g., calcite, dolomite) begins around 550 °C (Hazera et al., 2023), whereas some carbonates like siderite or oxalates start their decomposition around 520 °C (Lafargue et al., 1998). Reducing the final temperature of pyrolysis to 520 °C offers a practical compromise: it prevents the SIC decomposition during pyrolysis while still preserving a meaningful HC signal for characterizing soil OM. In this approach, SIC would be quantified exclusively during the subsequent oxidation phase.

Therefore, the aim of this study was to improve the direct quantification of SOC and SIC with RE by lowering the final temperature of pyrolysis rather than applying post-hoc corrections on TOC and MINC parameters. To this end, 173 soils and nine reference materials were analyzed with both the standard analysis cycle (noted PYRO650) and a cycle with a final pyrolysis temperature of 520 °C (noted PYRO520). The SOC and SIC quantifications using TOC and MINC parameters directly obtained with the two cycles (i.e., without post-hoc corrections) were compared to the SOC and SIC quantifications by EA after pretreatments. The indices characterizing OM obtained with the two cycles were compared. We hypothesized that (i) TOC and MINC parameters obtained with the PYRO520 cycle were respectively higher and lower than those obtained with the PYRO650 cycle, (ii) TOC and MINC parameters obtained with the PYRO520 would not differ significantly from the SOC and SIC contents estimated by EA, and (iii) the indices characterizing OM obtained with both cycles were linearly correlated.

## 2 Materials and methods

## 2.1 Materials

A total of 173 agricultural soils of varied geographical origins, land uses, and textures were selected (Fig. 1, Table S1). The soils were sampled at depths ranging from 0-10 cm to 210-220 cm (Table S1). Samples were sieved to 2 mm, milled with 200 μm mesh, and dried at 40 °C to constant weight before Rock-Eval® (RE) and elemental analyses (EA). The SOC contents measured by EA after removing SIC by acidification ranged from 1.2 g C kg<sup>-1</sup> to 68.2 g C kg<sup>-1</sup> and the SIC contents measured by EA after removing SOC by combustion ranged from 0.0 g C kg<sup>-1</sup> to 94.9 g C kg<sup>-1</sup> (Table S1). Among these soils, 26 were considered non-calcareous (SIC < 2.0 g C kg<sup>-1</sup>) and 147 were calcareous, with carbonates mainly in the form of calcite and, to a lesser extent, dolomite (Table S1).

Figure 1: Particle size distributions as the proportion of sand, silt, and clay (%) of the 173 soils. Cl: clay, SiCl: silty clay, SaCl: sandy clay, ClLo: clay loam, SiClLo: silty clay loam, SaClLo: sandy clay loam, Lo: loam, SiLo: silty loam, SaLo: sandy loam, Si: silt, LoSa: loamy sand, Sa: sand (USDA classification).

Nine reference materials from environmental agencies were also analyzed (Table 1). Their reference SOC contents ranged from 2.6 g C kg<sup>-1</sup> to 56.1 g C kg<sup>-1</sup> and their reference SIC contents ranged from 0.4 g C kg<sup>-1</sup> to 54.8 g C kg<sup>-1</sup> (Table 1).

5

Table 1: Name, type, environmental agency and TC, SOC, SIC reference contents (g C kg<sup>-1</sup>) of the nine reference materials

| Name          | Type                         | Environmental agency                                                                           | TC   | SOC  | SIC  |
|---------------|------------------------------|------------------------------------------------------------------------------------------------|------|------|------|
| SR-1          | Rock                         | Norwegian Petroleum Directorate                                                                | 36.9 | 22.4 | 14.5 |
| NIST-8704     | River sediment               | National Institute of Standards and Technology USA                                             | 33.5 | 24.0 | 9.1  |
| PACS-3        | Marine sediment              | National Research Council Canada                                                               | 32.9 | 31.7 | 1.2  |
| ERM-<br>CC690 | Calcareous soil              | Joint Research Centre Institute for Reference<br>Materials and Measurements                    | 93.0 | 56.1 | 36.9 |
| BCR-280R      | Lake sediment                | Joint Research Centre Institute Reference Materials and Measurements                           | 17.4 | 14.1 | 3.4  |
| ISE-850       | Calcareous soil              | Wageningen Evaluating Programs for Analytical<br>Laboratories                                  | 68.3 | 2.5  | 65.8 |
| AAFC-01       | Calcareous soil (eroded)     | Agriculture and Agri-Food Canada, Lethbridge (intern)                                          | 27.5 | 11.1 | 16.5 |
| AAFC-02       | Calcareous soil (non-eroded) | Agriculture and Agri-Food Canada, Lethbridge (intern)                                          | 20.9 | 13.6 | 7.2  |
| Till-3        | Subsoil                      | Natural Resources Canada (NRCan) Canadian Centre<br>for Mineral and Energy Technology (CANMET) | 11.5 | 11.1 | 0.4  |

# 135 **2.2 Elemental analysis**


Total carbon (TC), organic carbon (SOC), and inorganic carbon (SIC) contents were estimated on three aliquots using an elemental analyzer. The aliquots for TC quantification were analyzed directly by EA without pretreatment. Carbonates of the aliquots dedicated to SOC quantification were removed by acid treatment before EA (Table 2). SOC of the aliquots dedicated to SIC quantification was destroyed by combustion at 550 °C before EA (Table 2). The TC, SOC, and SIC contents estimated by EA are noted TC<sub>EA</sub>, SOC<sub>HCI+EA</sub>, and SIC<sub>550°C+EA</sub>, respectively.

Table 2: Sample weight (mg), device and protocol used by the four laboratories for SOC and SIC quantification by elemental analysis (EA) or thermal ramp

| Samples                | Weight                             | Device                                           | SOC protocol                                  | SIC protocol                                 |  |  |  |
|------------------------|------------------------------------|--------------------------------------------------|-----------------------------------------------|----------------------------------------------|--|--|--|
| France, Italy, Tunisia | 13-20                              | Elementar Vario                                  | Fumigation with HCl                           | Muffle furnace ignition at                   |  |  |  |
| (n = 113)              | 15 20                              | Isotope Select                                   | (12 M) + EA                                   | 550 °C (6 h) + EA                            |  |  |  |
| Germany (n = 15)       | 0.2-10<br>(SOC)<br>15-800<br>(SIC) | Thermo Flash 2000<br>EA (SOC)<br>LECO R612 (SIC) | Fumigation with HCl (12 M) + EA               | Thermal ramp analysis after 550 °C           |  |  |  |
| Hungary $(n = 30)$     | 2-4                                | Thermo Flash 2000<br>EA                          | Acidification HCl (3 M) + washing + EA        | Muffle furnace ignition at 550 °C (6 h) + EA |  |  |  |
| Canada (n = 15)        | 5-200                              | Thermo Flash 2000<br>EA                          | Small-scale acidification with HCl (6 M) + EA | Muffle furnace ignition at 550 °C (6 h) + EA |  |  |  |

Carbon contents were measured using EA and conducted in four laboratories. Sample weights, elemental analyzer devices, and pretreatment protocols differed among the four laboratories (Table 2). C content estimations by EA were replicated between one and five times for soil samples and at least four times for the reference materials. Each reference material was analyzed by the four laboratories (Table 2).

## 2.3 Rock-Eval® thermal analysis

## 150 2.3.1 Method description


The TC, SOC, and SIC contents were estimated on one untreated aliquot using a RE6Standard apparatus (Vinci Technologies, France). SOC and SIC contents were estimated by the TOC and MINC parameters, respectively. The TC content was estimated by the sum of the TOC and MINC parameters. Steel crucibles were filled with  $60 \pm 1$  mg of sample and analyzed using the standard "Bulk Rock" method. The analysis consists of a pyrolysis phase under an inert atmosphere (N<sub>2</sub>, purity = 99.99%) during which hydrocarbon compounds (HC), carbon monoxide (CO), and carbon dioxide (CO<sub>2</sub>) emissions are measured, followed by an oxidation phase under synthetic air (80% N<sub>2</sub> and 20% O<sub>2</sub>, purity = 99.99%) during which CO and CO<sub>2</sub> emissions are measured. Thus, five thermograms are obtained at the end of the analysis.

RE analyses were replicated three times for the reference materials as well as for 11 soil samples distributed across the TC content range of the studied soil samples (Table S1).


## 160 2.3.2 Calculation of the TOC and MINC parameters

Two RE cycles were performed on each sample with differing final pyrolysis temperatures: a standard pyrolysis with a final temperature of 650 °C (noted PYRO650), and a pyrolysis with a reduced final temperature of 520 °C (noted PYRO520). In both cycles, the pyrolysis started with an isotherm of 3 min at 200 °C and continued with a temperature ramp of 25 °C min<sup>-1</sup>. The PYRO650 cycle ended at 650 °C without an isotherm (Behar et al., 2001), whereas the PYRO520 cycle ended with an isotherm of 5 min at 520 °C to complete the thermal pyrolysis of OM decomposing at this temperature. The sample was cooled down at the end of the pyrolysis before the oxidation phase. The oxidation phase was identical for both cycles. The oxidation started with an isotherm of 3 min at 200 °C and continued with a temperature ramp of 25 °C min<sup>-1</sup> up to an isotherm of 7 min at 850 °C to ensure the complete thermal decomposition of carbonates (Hazera et al., 2023).

The five thermograms obtained with the PYRO650 cycle were divided into nine curves: S1 and S2 refer to the HC emitted during and after the 200 °C isotherm, respectively; S3CO and S3'CO display the CO emitted during pyrolysis before and after 550°C, respectively; S3CO<sub>2</sub> and S3'CO<sub>2</sub> denote the CO<sub>2</sub> emitted during pyrolysis before and after 550°C, respectively; S4CO refers to the CO emitted during oxidation, and S4CO<sub>2</sub> and S5, to the CO<sub>2</sub> emitted during oxidation before and after 650°C, respectively (Table 3).

Table 3: Temperature spans (°C) and contributions to the TOC and MINC parameters (%) of the curves obtained with the PYRO650 and PYRO520 cycles (n = 173)

| _                |               |         | PY        | OXIDATION  |                   |                    |              |                   |         |
|------------------|---------------|---------|-----------|------------|-------------------|--------------------|--------------|-------------------|---------|
|                  |               | НС      |           | CO         |                   | $CO_2$             |              | $CO_2$            |         |
|                  | S1            | S2      | S3CO      | S3'CO      | S3CO <sub>2</sub> | S3'CO <sub>2</sub> | S4CO         | S4CO <sub>2</sub> | S5      |
| PYRO650          |               |         | 1         |            |                   |                    |              |                   |         |
| Temperature span | 200           | 200-650 | 200-550   | 550-650    | 200-550           | 550-650            | 200-650      | 200-650           | 650-850 |
| Parameter        | TOC           | TOC     | TOC       | TOC/MINC   | TOC               | MINC               | TOC          | TOC               | MINC    |
| Contribution to  | the $0 \pm 0$ | 11 ± 5  | 2 ± 1     | $1 \pm 0/$ | 16 ± 7            | 13 ± 15            | 4 ± 1        | 66 ± 6            | 86 ± 17 |
| parameter        | 0 ± 0         | 11 ± 3  | $Z \pm 1$ | $2 \pm 3$  | 10 ± /            | 13 ± 13            | 4 _ 1        | 00 ± 0            | 00 ± 17 |
| PYRO520          |               |         |           |            |                   |                    |              |                   |         |
| Temperature span | 200           | 200-520 | 200-520   | -          | 200-520           | -                  | 200-850      | 200-650           | 650-850 |
| Parameter        | TOC           | TOC     | TOC       | -          | TOC               | -                  | TOC          | TOC               | MINC    |
| Contribution to  | the $0 \pm 0$ | 9 ± 5   | 2 ± 1     |            | $16 \pm 6$        | _                  | 4 ± 1        | $68 \pm 5$        | 100     |
| parameter        | 0 ± 0         | 9 = 3   | 2 ± 1     | -          | 10 ± 0            | -                  | <b>+</b> ± 1 | 00 ± 3            | 100     |

The parameters obtained with the PYRO650 cycle are noted TOC<sub>650</sub>, MINC<sub>650</sub> (Eqs 1 and 2), and TC<sub>650</sub>.

$$TOC_{650} = S1 + S2 + S3C0 + \frac{1}{2}S3'C0 + S3CO_2 + S4CO + S4CO_2$$
 (1)




$$MINC_{650} = \frac{1}{2}S3'CO + S3'CO_2 + S5$$
 (2)

The S3'CO and S3'CO<sub>2</sub> curves obtained with the PYRO520 cycle were null (Table 3). The parameters obtained with the PYRO520 cycle are noted TOC<sub>520</sub>, MINC<sub>520</sub> (Eqs 3 and 4), and TC<sub>520</sub>.

$$TOC_{520} = S1 + S2 + S3C0 + S3CO_2 + S4CO + S4CO_2$$
 (3)

$$MINC_{520} = S5 \tag{4}$$

The relative errors of the TOC and MINC parameters were estimated at 1.2 % and 3.3 % respectively (Pacini et al., 2023).

#### 2.3.3 Calculation of the indices characterizing organic matter

The main indices used in the literature to characterize soil OM are the hydrogen index (HI), the oxygen index (OIRE6), the I-index, and the R-index, which are directly calculated from the thermograms (e.g., Chassé et al., 2021; Deluz et al., 2024; Sebag et al., 2022). In addition, there is the proportion of stable SOC on a century scale (Cs) predicted by the Party<sub>SOC</sub> model (e.g., Cécillon et al., 2018, 2021; Delahaie et al., 2024, 2023).

The HI corresponds to the amount of HC emitted during pyrolysis relative to the TOC (Eq 5, mg HC g<sup>-1</sup> TOC; Behar et al., 2001; Carrie et al., 2012). The OIRE6 relates to the amount of oxygen emitted as CO and CO<sub>2</sub> during pyrolysis of OM relative to the TOC (Eq 6, mg O<sub>2</sub> g<sup>-1</sup> TOC; Behar et al., 2001; Carrie et al., 2012). The HI and OIRE6 are correlated with the H:C and O:C ratios of OM, respectively (Disnar et al., 2003). The HI = f(OIRE6) diagram is used to assess the decomposition state of OM: HI generally decreases with depth, reflecting the progressive oxidation of OM, i.e., dehydrogenation (decrease of HI) and relative oxidation (increase of OIRE6) of OM (Disnar et al., 2003; Barré et al., 2016; Sebag et al., 2022a). The HI and OIRE6 were obtained using the same calculation for both cycles of analysis but they are not equivalent as the temperature span of the S2 and S3CO<sub>2</sub> curves, as well as the TOC value, differ for each cycle (Table 3, Eqs 1 and 3). The HI and OIRE6 are noted HI<sub>650</sub>, OIRE6<sub>650</sub>, HI<sub>520</sub>, and OIRE6<sub>520</sub> when obtained with the PYRO650 and the PYRO520 cycles, respectively.

$$HI = \frac{S2}{TOC} \times 100 \tag{5}$$

OIRE6 = 
$$\frac{16}{28} \times \frac{\text{S3CO}}{\text{TOC}} \times 100 + \frac{32}{44} \times \frac{\text{S3CO}_2}{\text{TOC}} \times 100$$
 (6)

The I and R indices are calculated from a subdivision of the S2 curve into five areas: A1 (200-340 °C), corresponding to highly labile biopolymers, A2 (340-400 °C), labile biopolymers, A3 (400-460 °C), resistant biopolymers, A4 (460-520 °C), refractory biopolymers, and A5 (> 520 °C), highly refractory biopolymers (Sebag et al., 2016). The I and R indices correspond to the proportions of thermally labile and thermally stable OM, respectively, and are calculated from the A1, A2, A3, A4, and A5 areas expressed as percentages of the S2 curve (Eqs 7, 8, and 9). Since the A5 area does not apply with the PYRO520 cycle, the calculation of the R index was modified accordingly. The I and R indices are noted I<sub>650</sub> and R<sub>650</sub> (Eqs 7 and 8) when obtained with the PYRO520 cycle and I<sub>520</sub> and R<sub>520</sub> (Eqs 7 and 9) when obtained with the PYRO520 cycle.

$$I_{650} = \log_{10}[(A1 + A2)/A3] = I_{520}$$
 (7)

$$R_{650} = (A3 + A4 + A5)/100 (8)$$

$$R_{520} = (A3 + A4)/100 (9)$$

The I = f(R) diagram is used to investigate the dynamics of OM decomposition in the studied samples by comparing results with those from a conventional decomposition model, as described in Sebag et al. (2016).

The Cs proportion is estimated by the PARTY<sub>SOC</sub> model. The PARTY<sub>SOC</sub> model is a random forest machine learning model that estimates the Cs proportion of a sample from several RE parameters that have shown strong correlation with the Cs proportion of soils under bare fallow for several decades (Cécillon et al., 2018). The cycle used by Cécillon et al. (2018) was similar to the PYRO650 one but differed from it in its pyrolysis and oxidation temperature ramps: 30 °C min<sup>-1</sup> and 20 °C min<sup>-1</sup> respectively, versus 25 °C min<sup>-1</sup> for the PYRO650 cycle. Fifteen soil samples were analyzed with the cycle used by Cécillon et al. (2018) to assess the influence of these differences on the obtained Cs proportions (N° 9, 21, 23, 27, 58, 66, 220 69, 91, 92, 109, 151, 157, 168, 175, 177; Table S1). The Soil Carbon module of the Geoworks<sup>TM</sup> software (Geoworks V1.8R1, Vinci Technologies) was used to predict the Cs proportions for each soil sample and each RE analysis cycle with the PARTY<sub>SOC</sub> v2.0 model (Cécillon et al., 2021). With both the PYRO650 cycle and the cycle from Cécillon et al. (2018), the Geoworks<sup>TM</sup> software integrates CO and CO<sub>2</sub> pyrolysis signals up to 560 °C and the CO<sub>2</sub> oxidation signal up to 611 °C 225 (Cécillon et al., 2018). With the PYRO520 cycle, the HC, CO, and CO<sub>2</sub> pyrolysis signals were integrated across the whole pyrolysis temperature range. The Cs proportions obtained with the PYRO650, PYRO520 cycles, and the one used by Cécillon et al. (2018) are noted Cs<sub>650</sub>, Cs<sub>520</sub>, and Cs<sub>Cécillon</sub>, respectively.

### 2.4 Data analysis




The measurement repeatability was assessed using the median of standard deviations and coefficients of variation. The measurement accuracy was evaluated using the median of the absolute values of the relative error with respect to the reference contents of standard materials. For the measurements of the SIC content, only calcareous samples (SIC  $\geq$  2 g C kg<sup>-1</sup>) were included, specifically, this involved seven reference materials (Table 1), and 37 soil samples that were reanalyzed from the total of 147 calcareous soil samples.

The significance of the differences between the paired variables was evaluated within a 95% confidence interval using the Student's test for parametric variables (H0: the variables are not different, *t-test* function of the statistical software R, R Core Team, 2024) and the Wilcoxon's test for non-parametric variables (H0: the variables are not different, *wilcox.test* function of R). Ordinary least squares regressions between the variables were tested with the linear model fitting function (*lm* function of R) without their intercept. The overall significance of the regression was evaluated with the Fisher's test (H0: the relationship between the two variables is not significant). The coefficient of determination  $R^2$  reflects the proportion of variance explained by the regression. The significance of the difference between the regression slope and 1 was evaluated within a 95% confidence interval using the Student's test (H0: the slope is equal to 1, *t-test* function of R). The regression slopes are given with their standard errors.

#### 3 Results and discussion

#### 3.1 Quantification of soil organic and inorganic carbon with the PYRO520 cycle of the Rock-Eval® device

TC<sub>EA</sub> was well estimated by TC<sub>650</sub> (TC<sub>650</sub> =  $(1.00 \pm 0.00)$  TC<sub>EA</sub>, p-value (p) < 0.001, R<sup>2</sup> = 0.997; Fig. S1), while TOC<sub>650</sub> underestimated SOC<sub>HCl+EA</sub> by 9% (TOC<sub>650</sub> =  $(0.91 \pm 0.01)$  SOC<sub>HCl+EA</sub>; Fig. 2) and MINC<sub>650</sub> overestimated SIC<sub>550°C+EA</sub> by 2% (MINC<sub>650</sub> =  $(1.02 \pm 0.01)$  SIC<sub>550°C+EA</sub>; Fig. 2).

Figure 2: SOC and SIC contents (g C kg<sup>-1</sup>) estimated by TOC<sub>650</sub> and MINC<sub>650</sub> (gray) and TOC<sub>520</sub> and MINC<sub>520</sub> (black), respectively (n = 182 samples), vs SOC and SIC contents estimated by SOC<sub>HC1+EA</sub> and SIC<sub>550°C+EA</sub>, respectively (n = 173 soils), and reference SOC and SIC contents (n = 9 reference materials). Soil samples are represented by dots, and reference materials by triangles. Error bars show the standard deviations of SOC and SIC estimations by SOC<sub>HC1+EA</sub> and SIC<sub>550°C+EA</sub> (black symbols only, n = 117), TOC<sub>650</sub> and MINC<sub>650</sub> (n = 20 i.e., 11 soils + 9 reference materials), and TOC<sub>520</sub> and MINC<sub>520</sub> (n = 20), respectively. The 1:1 line (y = x) is plotted as a solid black line. Regression slopes are significantly different from zero (p-value (p) 

270

275

Figure 3: Distributions of deviations (g C kg $^{-1}$ ) between the SOC content estimated by TOC $_{650}$  (top) or TOC $_{520}$  (bottom) and by SOC $_{HCI+EA}$  or the reference SOC content (left) and between the SIC content estimated by MINC $_{650}$  (top) and MINC $_{520}$  (bottom) and by SIC $_{550^{\circ}C+EA}$  or the SIC reference content (right, n = 182). The zero-deviation line is plotted as a solid black line. The medians of the deviation are plotted as red dashed lines.

These results confirm that, with the PYRO650 cycle, a systematic error in the signal allocation between the TOC<sub>650</sub> and MINC<sub>650</sub> parameters is added to the random measurement error of the Rock-Eval® device. Delahaie et al., (2023), Hazera et al. (2023) and Stojanova et al. (2024) reported underestimations by the TOC parameter of 13%, 16%, and 8%, respectively, and overestimations by the MINC parameter of 7%, 4%, and 4%, respectively, with RE analysis cycles identical or similar to the PYRO650 cycle. The reported percentage of deviations between the EA and RE estimations are in the same range of values but vary depending on the datasets. These observations confirm that correcting these deviations with a fixed coefficient estimated on specific datasets could lead to substantial errors. Recently, this correction has been statistically modeled using machine learning algorithms to avoid errors related to the use of a fixed correction coefficient for all soils (Stojanova et al., 2024). However, this latter method was also calibrated on SOC and SIC contents estimated with the standard quantification methods (see Introduction), and only on French agricultural topsoils with a SOC content not exceeding 50 g C kg<sup>-1</sup> so far.

TC<sub>EA</sub> was also well estimated by TC<sub>520</sub> (TC<sub>520</sub> =  $(1.01 \pm 0.00)$  TC<sub>EA</sub>, p < 0.001, R<sup>2</sup> = 0.997; Fig. S1). TOC<sub>520</sub> was statistically higher than TOC<sub>650</sub> (Wilcoxon's test: p < 0.05) and was not statistically different from SOC<sub>HCl+EA</sub> (Wilcoxon's test: p = 0.96). The slope of the regression between SOC<sub>HCl+EA</sub> and TOC<sub>520</sub> (TOC<sub>520</sub> =  $(0.98 \pm 0.01)$  SOC<sub>HCl+EA</sub>; Fig. 2) was not statistically different from 1 (Student's test: p = 0.222, Fig. 2). The mean of the absolute deviations between TOC<sub>520</sub> and SOC<sub>HCl+EA</sub> was larger for samples with SOC content  $\geq 40$  g C kg<sup>-1</sup> (8.95  $\pm 5.14$  g C kg<sup>-1</sup>, n = 11) than for samples with SOC content < 40 g C kg<sup>-1</sup> (1.66  $\pm 1.78$  g C kg<sup>-1</sup>). Disnar et al. (2003) previously reported particularly large deviations between


the RE and EA estimations for "biopolymer-rich samples". MINC<sub>520</sub> was statistically lower than MINC<sub>650</sub> (Wilcoxon's test: p < 0.05) but remained statistically different from SIC<sub>550°C+EA</sub> (Wilcoxon's test:  $0.01 ). However, the slope of the regression between MINC<sub>520</sub> and SIC<sub>550°C+EA</sub> (MINC<sub>520</sub> = <math>(1.00 \pm 0.00)$  SIC<sub>550°C+EA</sub>; Fig. 2) was not significantly different from 1 (Student's test: p = 0.801, Fig. 2). The deviations between TOC<sub>520</sub> and SOC<sub>HCl+EA</sub>, and between MINC<sub>520</sub> and SIC<sub>550°C+EA</sub>, showed distributions centered around zero, comparable to the distribution of random measurement errors (Fig. 3).

The repeatability of RE analyses was satisfactory. The medians of the standard deviations and the coefficients of variation obtained by RE were lower than or equivalent to the ones obtained by EA (Table 4).

Table 4: Medians of standard deviation (SD, g C kg<sup>-1</sup>) and coefficient of variation (CV, %) of the total carbon (TC), organic carbon (SOC) and inorganic carbon (SIC) content estimations replicated at least 3 times on both the soil collection and the reference material sets. Median of the absolute relative errors (%) calculated with respect to the reference values given by the environmental agencies for TC, SOC and SIC content estimations. N indicates the number of samples replicated and n indicates the number of analyses used for the calculations. SIC\*: standard deviations, coefficients of variation and relative errors are reported for calcareous samples only (SIC content  $\geq$  2 g C kg<sup>-1</sup>).

| C form | Method                  | Soil collection set |     |     | Reference material set |                |     |      |   |     |
|--------|-------------------------|---------------------|-----|-----|------------------------|----------------|-----|------|---|-----|
|        |                         | SD                  | CV  | N   | n                      | Relative error | SD  | CV   | N | n   |
| SOC    | $SOC_{HCl+EA}$          | 0.4                 | 4.0 | 117 | 408                    | 11.7           | 2.1 | 12.7 | 9 | 283 |
|        | $TOC_{650}$             | 0.2                 | 1.7 | 11  | 33                     | 5.0            | 0.1 | 0.8  | 9 | 27  |
|        | $TOC_{520}$             | 0.2                 | 2.5 | 11  | 33                     | 8.7            | 0.1 | 0.4  | 9 | 27  |
|        |                         |                     |     |     |                        |                |     |      |   |     |
| SIC*   | $SIC_{550^{\circ}C+EA}$ | 0.3                 | 2.5 | 37  | 112                    | 21.9           | 1.7 | 14.6 | 7 | 152 |
|        | MINC <sub>650</sub>     | 0.5                 | 1.7 | 10  | 30                     | 13.8           | 0.2 | 1.5  | 7 | 21  |
|        | MINC <sub>520</sub>     | 0.5                 | 1.4 | 10  | 30                     | 5.1            | 0.1 | 1.1  | 7 | 21  |
|        |                         |                     |     |     |                        |                |     |      |   |     |
| TC     | $TC_{EA}$               | 0.4                 | 1.5 | 50  | 155                    | 2.9            | 1.3 | 4.0  | 9 | 278 |
|        | TC <sub>650</sub>       | 0.6                 | 1.3 | 11  | 33                     | 2.0            | 0.2 | 1.0  | 9 | 27  |
|        | $TC_{520}$              | 0.4                 | 1.0 | 11  | 33                     | 1.4            | 0.1 | 0.3  | 9 | 27  |

For the reference materials, the standard deviations and the coefficients of variation of EA were calculated from the results of four laboratories, which explained their higher values compared to the ones of soils and RE analyses. Nevertheless, RE analyses exhibited the lowest standard deviations and coefficients of variation among the five laboratories (data not shown). The medians of the standard deviations and the coefficients of variation obtained with the PYRO520 cycle were comparable to the ones obtained with the PYRO650 cycle (Table 4). The repeatability of the PYRO520 cycle was equivalent to that of EA or the PYRO650 cycle.



The values obtained by RE seemed more accurate than those obtained by EA. The medians of the absolute relative errors obtained by RE were lower than the ones obtained by EA (Table 4), yet the reference values were estimated by EA. Although the medians of relative errors for TC<sub>EA</sub> and SIC<sub>550°C+EA</sub> were calculated from the results of four laboratories, they are consistent with the ones reported by Shamrikova et al. (2023): between 10% and 23% for TC contents ranging from 1 g C kg<sup>-1</sup> to 300 g C kg<sup>-1</sup> and between 15% and 25% for SIC contents ranging from 1 g C kg<sup>-1</sup> to 120 g C kg<sup>-1</sup>. The relative errors obtained by RE were higher than the ones reported by Pacini et al. (2023) of 1.2% for TOC and 3.3% for MINC. However, the relative errors reported by Pacini et al. (2023) were calculated with respect to the mean value obtained by a set of RE devices and not with respect to a reference value. Thus, the relative errors reported by Pacini et al. (2023) reflect the inter-laboratory reproducibility of RE results rather than the accuracy of RE results. The relative errors obtained with the PYRO520 cycle were similar to the ones obtained with the PYRO650 cycle (Table 4). The accuracy of the estimations obtained with the PYRO520 cycle was finally satisfactory.

Thus, lowering the final temperature of pyrolysis at 520°C (i) corrected the misallocation of the end-of-pyrolysis signals in the TOC and MINC parameters without the need for post-hoc corrections, and (ii) improved the estimations of the SOC and SIC contents estimated by EA after acid or heating pretreatments. The TOC<sub>520</sub> and MINC<sub>520</sub> parameters were reproducible and accurate estimators of the SOC<sub>HCI+EA</sub> and SIC<sub>550°C+EA</sub> values.

# 3.2 Consequences of the change of the analysis cycle on the indices of organic matter characterization

#### 3.2.1 Hydrogen index and oxygen index

The hydrogen indices (HI) and oxygen indices (OIRE6) obtained with the PYRO520 and PYRO650 cycles were statistically different (Wilcoxon's tests: p < 0.05) but remained within similar ranges. HI<sub>650</sub> range was 20-310 mg HC g<sup>-1</sup> TOC and HI<sub>520</sub> range was 16-285 mg HC g<sup>-1</sup> TOC (Fig. 4a). OIRE6<sub>650</sub> range was 216-1159 mg O<sub>2</sub> g<sup>-1</sup> TOC and OIRE6<sub>520</sub> range was 256-1149 mg O<sub>2</sub> g<sup>-1</sup> TOC (Fig. 4b).



Figure 4: Comparison of the indices characterizing OM obtained with the PYRO650 and PYRO520 cycles (n = 173). a. HI<sub>520</sub> vs 325 HI<sub>650</sub> (mg HC g<sup>-1</sup> TOC) b. OIRE6<sub>520</sub> vs OIRE6<sub>650</sub> (mg O<sub>2</sub> g<sup>-1</sup> TOC) c. HI<sub>520</sub> =  $f(OIRE6_{520})$  in black and HI<sub>650</sub> =  $f(OIRE6_{650})$  in gray. d. I<sub>520</sub> vs I<sub>650</sub>. e. R<sub>520</sub> vs R<sub>650</sub>. f. I<sub>520</sub> =  $f(R_{520})$  in black and I<sub>650</sub> =  $f(R_{650})$  in gray. The 1:1 line (y = x) is plotted as a solid black line. Regression slopes are significantly different from zero (p-value (p) < 0.001). The p-values displayed on the graph indicate whether the slope significantly differs from 1. \* Slope significantly differs from 1.

Sebag et al. (2016) reported HI range of 5-400 mg HC  $g^{-1}$  TOC and OIRE6 range of 150- 1020 mg  $O_2 g^{-1}$  TOC for a diversity of A and B horizons (n = 527) analyzed with a cycle similar to the PYRO650 one. Delahaie et al. (2023) reported HI range of 67- 515 mg HC  $g^{-1}$  TOC and OIRE6 range of 75- 337 mg  $O_2 g^{-1}$  TOC for a diversity of French agricultural topsoils (n = 1891) analyzed with a cycle similar to the PYRO650 one. The HI and OIRE6 measured in this study were therefore consistent with those reported in the literature.

The HI and OIRE6 obtained with the PYRO520 cycle were linearly correlated with those obtained with the PYRO650 cycle ( $R^2 \ge 0.98$ , p < 0.001; Fig. 4a and 4b). HI<sub>520</sub> were on average lower than HI<sub>650</sub> (HI<sub>520</sub> = (0.83  $\pm$  0.01) HI<sub>650</sub>; Fig. 4a). The S2 curve integrations obtained with the PYRO520 cycle were reduced by the HC emitted between 520 °C and 650 °C thus their integrations were lower. As the TOC<sub>520</sub> parameter was higher than the TOC<sub>650</sub> parameter (Fig. 2), the S2:TOC ratio of the PYRO520 cycle used in the HI calculation (Eq. 5) was lower than the one of the PYRO650 cycle. OIRE6<sub>520</sub> were on average close to OIRE6<sub>650</sub> (OIRE6<sub>520</sub> = (0.99  $\pm$  0.01) OIRE6<sub>650</sub>; Fig. 4b). The reduction of the S3CO and S3CO<sub>2</sub> curves by the CO<sub>2</sub> emitted between 520°C and 550°C in the PYRO520 cycle was offset by the 5-minute isotherm at 520°C at the end of the






pyrolysis. The integrations of the S3CO and S3CO<sub>2</sub> curves obtained with the PYRO520 cycle were, on average, slightly higher than those obtained with the PYRO650 cycle (Fig. S2 and S3). As the TOC parameter was also higher with the PYRO520 cycle (Fig. 2), the S3CO:TOC and S3CO2:TOC ratios of the PYRO520 cycle used in the OIRE6 calculation (Eq. 6) were equivalent to those of the PYRO650 cycle. Only one soil sample (No. 168, Table S1) showed a large difference between OIRE6<sub>650</sub> and OIRE6<sub>520</sub>, respectively 1441  $\pm$  25 mg O<sub>2</sub> g<sup>-1</sup>TOC and 910  $\pm$  9 mg O<sub>2</sub> g<sup>-1</sup>TOC. This sample, which had a very low SOC content, showed the largest relative difference between TOC<sub>520</sub> (4.6  $\pm$  0.1 g C kg<sup>-1</sup>) and TOC<sub>650</sub> (2.1  $\pm$  0.0 g C kg<sup>-1</sup>), greatly influencing the S3CO:TOC and S3CO2:TOC ratios. Consequently, the OIRE6<sub>650</sub> of this sample was particularly high and aberrant compared to the OIRE6 values reported in the literature.

The general trend of the HI = f(OIRE6) diagram was preserved with the PYRO520 cycle (Fig. 4c). With the PYRO520 cycle, HI and OIRE6 are henceforth calculated with more accurate TOC values as they are no longer influenced by the presence of SIC. Moreover, the S2, S3CO, and S3CO2 curves are integrated over identical temperature intervals (200-520°C), contrary to HI<sub>650</sub> and OIRE6<sub>650</sub>, for which the S2 curve was integrated over 200-650 °C and the S3CO and S3CO<sub>2</sub> curves over 200-550 °C (Table 3). Consequently, HI<sub>520</sub> and OIRE6<sub>520</sub> are more coherent for characterizing organic matter in soil samples.

#### 3.2.2 I-index and R-index

I<sub>520</sub> and R<sub>520</sub> were statistically different from I<sub>650</sub> and R<sub>650</sub>, respectively (Student's test: p < 0.05), but remained within similar ranges. I<sub>650</sub> ranged from -0.35 to +0.40 and I<sub>520</sub> from -0.23 to +0.36. R<sub>650</sub> ranged from 0.44 to 0.84 and R<sub>520</sub> from 0.45 to 0.82. Sebag et al. (2016) reported I indices ranging from -0.18 to +0.64 and R indices from 0.29 to 0.80 for a diversity of A and B horizons (n = 527) analyzed with a cycle similar to the PYRO650 one. Delahaie et al. (2023) reported I indices ranging from -0.14 to +0.39 and R indices from 0.44 to 0.77 for a diversity of French agricultural topsoils (n = 1891) analyzed with a cycle similar to the PYRO650 one. The I and R indices measured in this study were therefore consistent with those reported in the literature.

The indices I and R obtained with the PYRO520 cycle were linearly correlated with those obtained with the PYRO650 cycle  $(R^2 \ge 0.9, p < 0.001; Fig. 4d and 4e)$ . I<sub>520</sub> were on average lower than I<sub>650</sub> (I<sub>520</sub> =  $(0.82 \pm 0.01)$  I<sub>650</sub>; Fig. 4d), whereas the R indices were on average not modified by the cycle change  $(R_{520} = (0.99 \pm 0.01) R_{650}; Fig. 6e)$ . This was due to the modification in the contributions of the A1 to A5 areas to the S2 curve used in the calculation of the I and R indices (Eq. 7, 8, and 9, Table 5).

Table 5: Proportions (mean ± standard deviation) of the A1, A2, A3, A4, and A5 areas (% of S2 curve) obtained with the PYRO650 and PYRO520 cycles. The p-values indicate whether the difference between the proportions obtained with the PYRO650 and PYRO520 cycles is statistically significant.

| -       | A1             | A2             | A3             | A4             | A5             |
|---------|----------------|----------------|----------------|----------------|----------------|
| PYRO650 | $12.3 \pm 5.0$ | $21.2 \pm 3.8$ | $28.8 \pm 2.9$ | $24.3 \pm 3.4$ | $13.3 \pm 4.9$ |
| PYRO520 | $12.2 \pm 4.6$ | $22.0 \pm 3.5$ | $30.6 \pm 2.8$ | $35.2 \pm 5.9$ | NA             |
| p-value | 0.25           | 



Figure 5: Proportion  $Cs_{520}$  vs proportion  $Cs_{650}$  (n = 173). The 1:1 line (y = x) is plotted as a solid black line and the  $Cs_{520}$  = 61 % and  $Cs_{520}$  = 78 % lines are plotted as dashed black lines. Regression slope is significantly different from zero (p-value (p) < 0.001). The p-value displayed on the graph indicates whether the slope significantly differs from 1. \* Slope significantly differs from 1.

The Cs proportion showed strong negative correlations with the parameters HI, S2, and the amount of pyrolyzed carbon (Cécillon et al., 2021). These parameters were lower with the PYRO520 cycle because a part of the OM that usually decomposes during pyrolysis was decomposed during the oxidation phase. Similarly, the Cs proportion showed positive correlations with the temperatures at which 50%, 70%, and 90% of the carbon is emitted as  $CO_2$  during the oxidation phase (Cécillon et al., 2021). These temperatures were higher with the PYRO520 cycle because the thermally stable fraction of OM that did not decompose during pyrolysis decomposed at higher temperatures during the oxidation phase. Thus, the Party<sub>SOC</sub> algorithm predicted higher Cs proportions with the signals obtained with the PYRO520 cycle than with those obtained with the PYRO650 cycle. The deviation of measurements between  $Cs_{650}$  and  $Cs_{520}$  seemed to depend on the magnitude of Cs value (Fig. 5). The deviations between  $Cs_{650}$  and  $Cs_{520}$  were larger for low Cs values: on average 6% for  $Cs_{520}$  values below 61% (n = 39), 1% for  $Cs_{520}$  values between 61% and 78% (n = 66), and 

#### 405 4 Conclusions




The TOC and MINC parameters obtained with the PYRO520 cycle were higher and lower than the ones obtained with the PYRO650 cycle, respectively. Lowering the final pyrolysis temperature corrected the misallocation of the end-of-pyrolysis signals between the TOC and MINC parameters and, thereby avoiding the need for post-hoc corrections. The TOC<sub>520</sub> and MINC<sub>520</sub> parameters estimated SOC and SIC contents measured by EA after pretreatments with both good repeatability and accuracy. The MINC<sub>520</sub> parameter provided accurate estimates of SIC contents up to  $100 \text{ g C kg}^{-1}$ . However, discrepancies between the TOC<sub>520</sub> parameter and the SOC<sub>HCI+EA</sub> values persisted for samples rich in SOC ( $\geq 40 \text{ g C kg}^{-1}$ ). Specific studies should focus on rich SOC samples to improve the estimation of their SOC contents by RE.

The modification of the pyrolysis phase decreased the HI and I indices while having almost no effect on the OIRE6 and R indices. Nevertheless, the general trends of the HI = f(OIRE6) and I = f(R) diagrams were preserved with the PYRO520 cycle. The proportions of Cs predicted by the Party<sub>SOC</sub> algorithm with the PYRO520 signals were higher than the ones

predicted with the PYRO650 signals, especially for soils with a predicted Cs<sub>520</sub> below 61%.

Since the HI, OIRE6, I, R indices, and the Cs proportions obtained with the two cycles were linearly correlated, the indices obtained with the PYRO550 cycle could be converted to be compared with those obtained with the PYRO520 cycle.

Data availability. The dataset is available at <a href="https://doi.org/10.57745/AL3NVT">https://doi.org/10.57745/AL3NVT</a>.

Supplement. SM1: Table S1, https://doi.org/10.57745/AL3NVT

SM2: Figures

Author contributions. JH, DS, TC, and IK designed the experiment. GJ, DZ, LT, RM, and JF sampled the soils. JH and HR performed the Rock-Eval measurements. GJ, DZ, FS, JF, and TC performed the elemental analysis. JH, TC, DS, IK, EV, and

425 HR analyzed the data. JH wrote the manuscript. TC, DS, IK, EV, GJ, DZ, FS, LT, RM and, JF reviewed and edited the manuscript.

Competing interests. The authors declare that they have no conflict of interest.

Acknowledgements. The authors thank Kirsten Hannam, Sandra Yanni and Benjamin Ellert for providing the soil samples from Canada and their elemental analyses.

The authors thank Nadhem Brahim, Patrice Coll, Edith LeCadre, Michel Brossard, and Lydie Lardy for soil collection and soil classification in Tunisia and France.

The authors thank Nathalie Crozet and Eric Kohler for performing the XRD analysis of the samples used in this study. Rock-Eval® is a trademark registered by IFP Energies Nouvelles.

Financial support. This research has been supported by the SIC-SOC-DYN "Organic and inorganic carbon dynamic in calcareous soils" project of the first external call "Towards Healthy, Resilient and Sustainable Agricultural Soils" within the

EJP SOIL program (2022–2025), Horizon 2020 (grant no. 862695).

The analyses of the Hungarian samples have been supported by the project no. 2019-2.14-ERA-NET-2022-00037 implemented with the support provided by the Ministry of Culture and Innovation of Hungary from the National Research, Development and Innovation Fund, financed under the ERA-NET COFUND/EJP COFUND funding scheme with cofunding from the European Union Horizon 2020 research and innovation program.

Joséphine Hazera was partly funded by the project ALAMOD of the exploratory research program FairCarboN and received government funding managed by the Agence Nationale de la Recherche under the France 2030 program, reference ANR-22-PEXF-0002 – project ALAMOD.

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
