# Peer review of "Technical note: Further adjustments to the Rock-Eval® thermal analysis for soil organic and inorganic carbon quantification to avoid post-hoc corrections"

_EGUsphere, 2025_

## Author Comment (AC1)

**Please note that our answers are in bold and that the line numbers correspond to the ones of the Ms with track change. The quotations in italics are the modified parts of the Ms.**

In the present manuscript, Hazera and co-authors present an extensive, well-characterized dataset of 173 soil samples and 9 reference materials analyzed with the established Rock-Eval method. They propose an adjustment to the standard temperature windows of this method and convincingly show that reducing the maximum pyrolysis temperature from 650°C down to 520°C can help reduce overlap between low-temperature carbonate phases and organic carbon signals at the end of pyrolysis and remove the need for post-hoc corrections which can induce bias. They show that most characteristic indices calculated using established protocols remain statistically comparable between the two methods and deviations are predictable. The only drawback of the adjusted method appears to be larger uncertainty at higher SOC values (upward of ~4 wt%), which represents a potentially significant limitation and requires further exploration in future work.

Overall, I commend the authors on a very clearly written and structured technical note which cites an extensive amount of current literature and firmly roots its claims in both previous work and statistical analysis. Potential problems of inter-lab comparability are well addressed by measuring common standards as well as a subset of samples across labs.

**The authors would like to thank the Anonymous Referee #1 for his/her attentive reading, his/her compliments on our study and his/her constructive comments. Indeed, we did our best to make this article understandable for non-Rock-Eval users and convincing by using reference materials and performing inter-lab comparison. We agree on all your comments, see our detailed response below.**

However, a few points require clarification and are listed below:

L47: Consider changing to "consists of"

**We changed to "*It consists of high-temperature combustion…*" (L47).**

L55-56: Add the mineral names calcite and dolomite after the chemical descriptions, as this is how you will refer to them throughout the rest of the text

**We added calcite and dolomite as examples in the sentence: "*… quantifying calcium carbonate (e.g., calcite) and thus misses other carbonate forms (e.g., calcium magnesium carbonate like dolomite)*" (L55-56) as calcium carbonate can also be in forms of aragonite and magnesium carbonate in forms of magnesite for instance.**

L61 (and many more occasions): Change the hyphen ("-") to an en-dash ("–") when giving ranges

**We changed at L61-62, L90, L123, Table 2, Table 3, L212-213, L350-361, and L382-384.**

L68-69: Change to: SIC "generally" decomposes[...], as you will later state how not all inorganic carbon is more temperature resistant than all organic carbon.

**We modified the sentence as advised: "*Since SIC generally decomposes…*" (L68).**

L72: Spell out Mineral Carbon (MINC) at least once

**We added the meaning of the two parameters: *"… named TOC (total organic carbon) and MINC (mineral carbon), respectively."* (L72).**

L100-103: At 520°C the decomposition of (admittedly pretty rare) siderite starts, so the present method does not really rule out an artifact from this. It should be briefly mentioned later in the discussion. As it is one of the central arguments of this procedure, I suggest being very precise around it.

**We preferred to remove the mention of siderite in the 1. Introduction section which was indeed confusing. The choice of 520 °C was rather motivated by the previous use of this temperature boundary in RE literature on soil so we specified it: *"With this standard pyrolysis, the temperature boundary of 520 °C is often used in soil literature to distinguish the HC signal emitted by refractory OM from the HC signal emitted by highly refractory OM (Malou et al., 2020; Deluz et al., 2024)."* (L96-98). Then we specified clearly that calcite and dolomite are the most common soil carbonates and that reducing the temperature at 520 °C prevent the decomposition of these carbonates only: *"Calcite and dolomite, the most common soil carbonates (Zamanian et al., 2016; Khalidy et al., 2022), begins their thermal decomposition around 550 °C during pyrolysis (Hazera et al., 2023). Reducing the final temperature of pyrolysis to 520 °C offers a practical compromise: it prevents the decomposition of calcite and dolomite during pyrolysis while still preserving a meaningful HC signal for characterizing soil OM."* (L103-109). We also specified in the 2.1 Materials section that XRD analyses were performed on the soil samples (see answer of the next comment) and summarized their results which can be found in the Table S1: *"The X-ray diffraction analyses detected quartz, phyllosilicates, and carbonates. Carbonates were mainly calcite and, to a lesser extent, dolomite (Table S1)."* (L127-129). Then, we specified in the 3.1 section that no siderite was detected by XRD in the soil samples and that this type of carbonate might impact the SOC and SIC quantification by RE: *"MINC$_{520}$ estimated properly SIC$_{550°C+EA}$ up to 100 g C kg$^{-1}$ for the samples of this study containing exclusively calcite and dolomite. No siderite or oxalate carbonates were detected by the X-ray diffraction analyses on the samples used in this study (Table S1). The thermal decomposition of siderite or oxalate carbonates can start before 400 °C during the pyrolysis phase (Ferro, 2012; Sebag et al., 2018) and thus might impact the SOC and SIC quantification by RE."* (L309-312). Finally, we added as a perspective in the 4. Conclusions section the need for specific studies for samples containing other carbonates than calcite or dolomite: *"Even if calcite, and dolomite are the main carbonates in soil, further investigations are needed on soil samples containing carbonates with a thermal breakdown beginning before 520 °C, e.g. siderite and oxalate, to assess the influence of these carbonates on the SIC quantification by RE."* (L441-444). We hope these changes clarify the choice of 520 °C and the applications of this method.**

L122-123: You mention the mineralogy of the samples. Table S1 is not (yet?) available under the doi, but I wonder if you quantified siderite or other carbonate content? It would help to add half a sentence about this, as the reader just learned (L100) that this is something to look out for.

**We hoped the revision would have been quicker but unfortunately the Table S1 is still under revision. We apologize for this inconvenience. We asked a private link that you can check the Table S1 before its publication:**

https://entrepot.recherche.data.gouv.fr/privateurl.xhtml?token=36591ec5-cc13-4f84-9fb5-a0d249c917cd. As mentioned above, no siderite or other "rare" carbonates were detected by XRD. Only calcite, dolomite, and traces of aragonite (another form of $CaCO_3$ as mentioned above) on 2 samples were detected.

L137: If you re-introduce the abbreviations, also reintroduce Elemental Analyzer (EA)

**The reintroductions of the abbreviations in the 2. Materials and methods section were indeed misplaced. We reintroduced SOC and SIC sooner (L124-125), after the reintroduction of EA and kept the reintroduction of TC at its first occurrence (L143). We also reintroduced all the abbreviations in the captions of Table 1 and Table 2, OM at its first occurrence (L145) and TOC and MINC at their first occurrence (L159-160).**

L138-139 and Table 2: Organic C from the samples was removed by heating to 550°C, although we learned earlier that even calcite starts to break down at this temperature. This raises the suspicion of a slight circularity in the argumentation. I assume a good match between TC measured by EA of untreated sample and SOC(HCl)+ SIC(550°) measured from the two treatments shows this was not a big problem, but maybe say that explicitly? The paper really focuses on these thermal boundary regions.

**The heating at 550 °C is a usual procedure to destroy SOC prior to EA or for SOC quantification by loss-on-ignition. However, the temperature reported in the literature to do so are variable (~450-600 °C, see L63-64). Using lower temperature can lead to incomplete SOC removal before EA… We are aware that these procedures can lead to discrepancies and errors in the results, this is why we wanted to promote thermal analyses which use only one unpretreated aliquot and allow the operator to choose the most appropriate temperature boundary. However, we had to compare our results with standard procedures to convince those who use these procedures that the RE results are somewhat consistent with those of the standard procedures. We added a discussion on the inconsistencies of EA results when $SOC_{HCl+EA}$ is compared to $TC_{EA} - SIC_{550°C+EA}$ (see answer to your comment on L277-279).**

Table 2: What is the difference between "acidification" and "small-scale acidification"? Was a smaller quantity of acid or material used? Consider removing this distinction if not relevant.

**The difference between acidification and small-scale acidification is the acid concentration (3 M vs 6 M) and the presence or absence of washing. It is indeed mentioned in the Table 2 thus we replaced "*small-scale acidification*" by "*acidification*" as suggested.**

L151-152: Consider rephrasing/combining the two sentences to make clear what was measured (TOC and MINC) and what was estimated based on the measurements (TC, SOC, SIC).

**We wanted to keep the same phrasing for EA and RE to make the link clearer thus we modified the sentences as follows: "*Three aliquots of each sample were analyzed using an elemental analyzer. The aliquots dedicated to total carbon (TC) quantification were analyzed directly by EA without pretreatment. Carbonates of the aliquots dedicated to SOC quantification were removed by acid treatment before EA (Table 2). Organic matters (OM) of the aliquots dedicated to SIC quantification were removed by combustion at 550°C before EA (Table 2).*" (L142-146) and "*One unpretreated aliquot of each sample was analyzed using a RE6Standard apparatus (Vinci Technologies, France). SOC and SIC contents were***

*estimated by the TOC (total organic carbon) and MINC (mineral carbon) parameters respectively. TC content was estimated by the sum of the TOC and MINC parameters."* **(L158-161).**

L158: Consider giving the TC range of samples used in the inter-lab comparison, similar to giving the range of standards used earlier.

**The reference materials cited here are the same than those described in Table 1, where their TC content is given, and are the ones used for the inter-lab comparison. As it seems unclear, we added the number of reference materials in the sentence:** *"…for the nine reference materials as well as for 11 soil samples distributed across the TC content range of the studied soil samples (Table S1)."* **(L166).**

L165: How was the 5-minute hold temperature determined? If no tests were conducted, cite a reference where this was shown.

**Usually there is no isotherm at the end of the pyrolysis, as in the PYRO650 cycle, to avoid further SIC decomposition during this phase. With the PYRO520 cycle, the SIC decomposition is avoided during the pyrolysis phase. Our aim was to preserve as much as possible the signals emitted by the pyrolysis of OM. The 5-min isotherm was chosen to properly complete the pyrolysis of OM decomposing at 520°C. First tests with a 3-min isotherm showed a sharply drop of the $CO_2$ signal at the end of the isotherm which will not impact the C quantification but will impact the OI calculation and thus the OM characterization. Therefore, we decided to extend this isotherm to 5 min and to not show these results to shorten the article. To clarify this decision, we modified the sentence as follows:** *"The PYRO650 cycle ended at 650 °C without an isotherm to avoid further SIC decomposition (Behar et al., 2001), whereas the PYRO520 cycle ended with an isotherm of 5 min at 520 °C to complete the thermal pyrolysis of OM decomposing at this temperature. An isotherm of 3 min was tested but seemed too short to properly estimate the amount of OM pyrolyzed at 520 °C (data not shown)."* **(L172-175).**

L188 (and multiple occasions): Consider referring to the century-stable carbon as C_S with the S in subscript for readability

**We changed Cs by $C_S$ where needed in 2.3.3, 3.2.3 and 4. sections as well as in the figures 5 and S4 and their captions.**

L220-221: Consider removing the dedicated sample numbers for the samples analyzed with PARTY_SOC, as we did not see sample numbers before, e.g., for the inter-lab samples. You can indicate the samples in Table S1.

**We removed the sample numbers and kept only the reference to the Table S1:** *"Fifteen soil samples were analyzed with the cycle used by Cécillon et al. (2018) to assess the influence of these differences on the obtained $C_S$ proportions (Table S1)."* **(L230-232) The sample numbers will be added in the data base description of the Table S1.**

L250-251: I assume the 182 samples are the 173 soils plus the 9 standards, but the phrasing makes it a bit ambigous if it's two sets of samples. Consider clarifying

**Indeed, the 182 samples are the 173 soils plus the 9 reference materials. We added the description of the 182 samples to clarify this point: "*(n = 182 i.e., 173 soils + 9 reference materials)*" (L261)**

L277-279: The performance of the PYRO520 method seems to be considerably worse for samples with more than 4 wt% OC, which is a value regularly exceeded in real-world samples. Maybe you can spend a few sentences on the reasons and consequences for the applicability of the modified procedure.

**We followed your idea of comparing the results obtained by EA. We compared the SOC$_{HCl+EA}$ values, and the SOC contents estimated as TC$_{EA}$ - SIC$_{550°C+EA}$ to discuss the performance of the PYRO520 cycle for SOC quantification. As expected, both ways to estimate SOC content by EA are not consistent. We added a discussion in the 3.1 section on the validity of the standard procedure itself: "*The larger mean of absolute deviations between TOC$_{520}$ et SOC$_{HCl+EA}$ for soil samples rich in SOC was consistent with the results of Disnar et al. (2003) who reported particularly large deviations between the RE and EA estimations for "biopolymer-rich samples". However, the high SOC content estimations by EA showed also inconsistencies. The mean of the absolute deviation between SOC$_{HCl+EA}$ and the SOC content estimated as TC$_{EA}$ - SIC$_{550°C+EA}$ was larger for samples with SOC content ≥ 40 g C kg$^{-1}$ (8.53 ± 7.71 g C kg$^{-1}$, n = 11) than for samples with SOC content < 40 g C kg$^{-1}$ (2.00 ± 2.22 g C kg$^{-1}$; Table S1). Several studies reported effects of acid pretreatment on the OM and/or incomplete SIC removal by acid pretreatment that can lead to substantial errors in SOC quantification (Schlacher and Connolly, 2014; Apesteguia et al., 2018; He et al., 2025). Moreover, Apesteguia et al. (2018) pointed out that subtracting carbon contents estimated on different aliquots can lead to inconsistencies in the results. Thus, the remaining discrepancies between SOC$_{HCl+EA}$ and TOC$_{520}$ could be due to underestimation of high SOC contents by RE or analytical errors in the standard procedure itself.*" (L296-306). We summarized and added perspectives on this discussion in the 4. Conclusions section: "*Discrepancies between the TOC$_{520}$ parameter and the SOC$_{HCl+EA}$ values persisted for samples rich in SOC (≥ 40 g C kg$^{-1}$). However, it is not straightforward to attribute these discrepancies to either RE analysis or EA analysis on pretreated aliquots. Specific studies should focus on non-calcareous soils rich in SOC to conclude on the RE ability to estimate high SOC content.*" (L444-447).**

L339: Consider rephrasing the sentence and maybe talk about the "absence" of the 520 to 550°C temperature range instead of a signal reduction, as these temperatures were never realized in PYRO520.

**We modified the sentence as suggested: "*The absence of CO and CO$_2$ signals between 520°C and 550°C in the PYRO520 cycle was offset by the 5-minute isotherm at 520°C at the end of the pyrolysis.*" (L369-370).**

L405 onward: In my opinion, another one or two sentences on ways forward and the general implications of the technical note are needed. The data is presented nicely and the most important results are repeated in the conclusion. But is this method now ready to be applied for low-TOC soil samples, or should higher-TOC samples, or a refinement of the HI, OIRE6 indices etc. be the next priority?

We added several perspectives in the conclusions: assessing the influence of rare carbonates on SIC quantification with the PYRO520 cycle (see answer to your comment on L100-103), assessing the capacity of RE analysis to estimate high SOC contents (see answer to your comment on L277-279) and performing a specific study on PartySOC utilization with the PYRO520 results ("*As the PartySOC algorithm is a machine learning model, a specific study should focus on the validation of the correlation presented in this article. Ideally, the PartySOC model could be recalibrated with data obtained with the PYRO520 cycle.*", L454-455).

---

## Author Comment (AC2)

**Please note that our answers are in bold and that the line numbers correspond to the ones of the Ms with track change. The quotations in italics are the modified parts of the Ms.**

This is a very nice and timely study. The idea to lower the pyrolysis temperature to 520°C is simple but brilliant, effectively solving the long-standing problem of signal overlap between SOC and SIC at the end of the standard pyrolysis. The manuscript is well-written, the dataset is extensive, and the results are convincing. This work has significant practical implications for improving Rock-Eval standard procedures.

**The authors would like to thank the Anonymous Referee #2 for his/her attentive reading, his/her compliments on our study and his/her constructive comments. Indeed, simplest solutions are sometimes the best. We agree on all your comments, see our detailed response below.**

However, a few points require clarification and are listed below:

Major Comments:

Did you test other temperatures (e.g., 515°C, 525°C)? A graph showing the SIC signal (from the oxidation phase) versus final pyrolysis temperature for a few samples would be very helpful to prove that 520°C is the optimal cutoff. Please provide a more detailed discussion on the decomposition temperatures of different carbonate minerals (calcite, dolomite, siderite) to better explain why 520°C is the chosen "practical compromise."

**We tested a final temperature pyrolysis of 550 °C on few soil samples and the reference materials and concluded that 550 °C was not low enough to avoid SIC thermal breakdown during the pyrolysis phase. We decided to not show these results to shorten the article. The choice of 520 °C was rather motivated by the previous use of this temperature boundary in RE literature on soil so we specified it in the 1. Introduction section: "*With this standard pyrolysis, the temperature boundary of 520 °C is often used in soil literature to distinguish the HC signal emitted by refractory OM from the HC signal emitted by highly refractory OM (Malou et al., 2020; Deluz et al., 2024).*" (L96-98). Using a lower temperature boundary than 520 °C could slightly improve the separation of the SOC and SIC pyrolysis signals but will lead to a loss of meaningful HC signal for OM characterization. The decomposition temperature of calcite and dolomite is given in the 1. Introduction section. To avoid any confusion, we preferred to remove the mention of siderite in the 1. Introduction section and specify clearly that calcite and dolomite are the most common soil carbonates and that reducing the temperature at 520 °C prevent the decomposition of these carbonates only: "*During pyrolysis, the thermal decomposition of common soil carbonates (e.g., calcite, dolomite) begins around 550 °C (Hazera et al., 2023). Reducing the final temperature of pyrolysis to 520 °C offers a practical compromise: it prevents the decomposition of calcite and dolomite during pyrolysis while still preserving a meaningful HC signal for characterizing soil OM.*" (L103-109). We specified in the 3.1 section that no siderite was detected by XRD (see answer of your next comment) in the soil samples and that this type of carbonate might impact the SOC and SIC quantification by RE because of their lower thermal breakdown temperature: "*$MINC_{520}$ estimated properly $SIC_{550°C+EA}$ up to 100 g C kg$^{-1}$ for the samples of this study containing exclusively calcite and dolomite. No siderite or oxalate carbonates were detected by the X-ray diffraction analyses on the samples used in this study (Table S1). The thermal decomposition of siderite***

*or oxalate carbonates can start before 400 °C during the pyrolysis phase (Ferro, 2012; Sebag et al., 2018) and thus might impact the SOC and SIC quantification by RE.*" (L309-312). **Then, we added as a perspective in the 4. Conclusions section the need for specific studies for samples containing other carbonates than calcite or dolomite:** "*Even if calcite, and dolomite are the main carbonates in soil, further investigations are needed on soil samples containing carbonates with a thermal breakdown beginning before 520 °C, e.g. siderite and oxalate, to assess the influence of these carbonates on the SIC quantification by RE.*" (L441-444). **We hope these changes clarify the choice of 520 °C and the applications of this method.**

The type of carbonate (e.g., calcite vs. dolomite) and its association with organic matter could affect its decomposition temperature. I recommend adding mineralogical data (e.g., XRD analysis) for a subset of samples to characterize the dominant carbonate phases.

**We hoped the revision would have been quicker but unfortunately the Table S1 is still under revision. We apologize for this inconvenience. We asked a private link that you can check the Table S1 before its publication: https://entrepot.recherche.data.gouv.fr/privateurl.xhtml?token=36591ec5-cc13-4f84-9fb5-a0d249c917cd. As mentioned above, no siderite or other "rare" carbonates were detected by XRD. We specified it in the 2.1 Materials section:** "*The X-ray diffraction analyses detected quartz, phyllosilicates, and carbonates. Carbonates were mainly calcite and, to a lesser extent, dolomite (Table S1).*" **(L127-129).**

Please discuss if the diagenetic carbonates in rocks behaved differently from the pedogenic carbonates in soils in your analyses. This will help clarify the applicability of your method to different sample types.

**We did not discuss the carbonate forms in this study to shorten the article. The isotopic signatures ($\delta^{13}C$) of the SIC of the soil samples will be add in the Table S1. $\delta^{13}C_{SIC}$ were variable enough (between 0 ‰ and -15 ‰) to assume the presence of lithogenic and pedogenic carbonates in our soil samples. Moreover, several reference materials were rock or sediment samples. We reminded this at the end of the 3.1 section:** "*The $TOC_{520}$ and $MINC_{520}$ parameters were reproducible and accurate estimators of the $SOC_{HCl+EA}$ and $SIC_{550°C+EA}$ values for the soil samples as well as for the reference materials including rock and sediment samples.*" **(L344-346). However, the PYRO520 is recommended mainly for soil applications because rocks and sediments can contain OM that are more thermos-stable than OM of soils. Thus, even if the organic and inorganic carbon quantification will be correct, the OM characterization of rocks and sediments could be affected by the reduction of the pyrolysis phase. This is why the initial end-of-pyrolysis temperature was set at 650 °C as mentioned in the 1. Introduction section (L94-96).**

Minor Comments:

The larger errors for samples with SOC $\geq$ 40 g C kg$^{-1}$ are noted. Please add a brief hypothesis for this observation (e.g., related to more refractory organic compounds like pyrogenic carbon?).

**We compared the $SOC_{HCl+EA}$ values and the SOC contents estimated as $TC_{EA}$ - $SIC_{550°C+EA}$ to discuss the performance of the PYRO520 cycle for SOC quantification. As expected, both ways to estimate SOC content by EA are not consistent. We added a discussion in the**

**3.1 section on the validity of the standard procedure itself:** " *The larger mean of absolute deviations between $TOC_{520}$ et $SOC_{HCl+EA}$ for soil samples rich in SOC was consistent with the results of Disnar et al. (2003) who reported particularly large deviations between the RE and EA estimations for "biopolymer-rich samples". However, the high SOC content estimations by EA showed also inconsistencies. The mean of the absolute deviation between $SOC_{HCl+EA}$ and the SOC content estimated as $TC_{EA}$ - $SIC_{550°C+EA}$ was larger for samples with SOC content $\geq 40$ g C $kg^{-1}$ (8.53 ± 7.71 g C $kg^{-1}$, n = 11) than for samples with SOC content < 40 g C $kg^{-1}$ (2.00 ± 2.22 g C $kg^{-1}$; Table S1). Several studies reported effects of acid pretreatment on the OM and/or incomplete SIC removal by acid pretreatment that can lead to substantial errors in SOC quantification (Schlacher and Connolly, 2014; Apesteguia et al., 2018; He et al., 2025). Moreover, Apesteguia et al. (2018) pointed out that subtracting carbon contents estimated on different aliquots can lead to inconsistencies in the results. Thus, the remaining discrepancies between $SOC_{HCl+EA}$ and $TOC_{520}$ could be due to underestimation of high SOC contents by RE or analytical errors in the standard procedure itself.*" (L296-306). **We summarized and added perspectives on this discussion in the 4. Conclusions section:** "*Discrepancies between the $TOC_{520}$ parameter and the $SOC_{HCl+EA}$ values persisted for samples rich in SOC ($\geq 40$ g C $kg^{-1}$). However, it is not straightforward to attribute these discrepancies to either RE analysis or EA analysis on pretreated aliquots. Specific studies should focus on non-calcareous soils rich in SOC to conclude on the RE ability to estimate high SOC content.*" (L444-447).

The correlation between Cs650 and Cs520 is excellent. A short comment on the model's transferability to the new PYRO520 thermal program would be useful for readers.

**The correlation is excellent, but it is not straightforward to understand why the deviation differ depending on the proportion estimated. Actually, we are not sure that the correlation between these predicted indices can be used like the correlations with the calculated indices thus we preferred to remove $C_S$ proportions of the last sentence and to add two sentences on what can be done concerning the $C_S$ proportions:** "*Since the HI, OIRE6, I, and R indices obtained with the two cycles were linearly correlated, the indices obtained with the PYRO650 cycle could be converted to be compared with those obtained with the PYRO520 cycle. As the PartySOC algorithm is a machine learning model, a specific study should focus on the validation of the correlation presented in this article. Ideally, the PartySOC model could be recalibrated with data obtained with the PYRO520 cycle.*" (L452-455).